# Wolf–Parkinson–White Syndrome: Diagnosis, Risk Assessment, and Therapy—An Update

**DOI:** 10.3390/diagnostics14030296

**Published:** 2024-01-30

**Authors:** Radu Gabriel Vătășescu, Cosmina Steliana Paja, Ioana Șuș, Simona Cainap, Ștefana María Moisa, Eliza Elena Cinteză

**Affiliations:** 1Cardiology Department, Clinic Emergency Hospital, 014461 Bucharest, Romania; cosminasteliana@gmail.com; 24th Department—Cardio-Thoracic Pathology, University of Medicine and Pharmacy “Carol Davila”, 020021 Bucharest, Romania; eliza.cinteza@umfcd.ro; 3Emergency Institute for Cardiovascular Disease and Transplantation, 540136 Tirgu Mures, Romania; susioana@yahoo.com; 48th Department—“Mother and Child”, University of Medicine and Pharmacy “Iuliu Hațieganu”, 400012 Cluj-Napoca, Romania; sorana.cainap@umfcluj.ro; 52nd Pediatric Department, Clinical Children Hospital, 400177 Cluj-Napoca, Romania; 6Department of Pediatrics, Faculty of Medicine, “Gr. T. Popa” University of Medicine and Pharmacy, 700115 Iasi, Romania; 7“Sfanta Maria” Clinical Emergency Hospital for Children, 700309 Iasi, Romania; 8Interventional Cardiology Compartment, Marie Sklodowska Curie Children Emergency Hospital, 077120 Bucharest, Romania

**Keywords:** accessory pathway, tachyarrhythmias, sudden cardiac death, risk assessment, radiofrequency ablation

## Abstract

Wolf–Parkinson–White (WPW) syndrome is a disorder characterized by the presence of at least one accessory pathway (AP) that can predispose people to atrial/ventricular tachyarrhythmias and even sudden cardiac death. It is the second most common cause of paroxysmal supraventricular tachycardia in most parts of the world, affecting about 0.1–0.3% of the general population. Most patients with WPW syndrome have normal anatomy, but it may be associated with concomitant congenital heart disease or systemic diseases. Although many individuals are asymptomatic, during supraventricular arrhythmia episodes, they may experience severe symptoms, including syncope or even sudden cardiac death (mainly due to pre-excited atrial fibrillation over rapidly conducting AP). In addition to arrhythmia-related symptoms, for some specific locations of the APs with overt anterograde conduction, there might be a reduction in exercise capacity mediated by a reduction in LV systolic performance due to anomalous LV depolarization. Although it is typically diagnosed through electrocardiography (ECG), additional tests are necessary for risk assessment. Management of WPW syndrome may be quite challenging and can vary from only acknowledging the presence of the accessory pathway to pharmacological treatment or radiofrequency ablation. Early diagnosis, risk assessment, and appropriate treatment are critical steps in the management of WPW syndrome, aiming to improve the quality of life and reduce the risk of life-threatening arrhythmias.

## 1. Introduction

In the 1970s–1980s, Wolff–Parkinson–White (WPW) syndrome was the main subject of several studies involving its epidemiology and pathogenesis. Electrophysiologists’ attention was drawn to this pathology, not only because of its possible clinical implications but also because of the opportunity for a more thorough study of the electrical proprieties of the heart tissue beyond the already known conduction system.

Management of patients with WPW syndrome can vary, from only acknowledging the presence of the accessory pathway to pharmacological treatment or radiofrequency ablation. Most of the patients, despite having ECG characteristics compatible with Wolff–Parkinson–White syndrome, never experience any symptoms, and, therefore, medical intervention is not required except in a few situations. On the other hand, in those patients who experience symptoms and in which tachyarrhythmias are documented, it is mandatory to assess the risk of sudden cardiac death and to establish the most adequate therapy.

## 2. Definitions and Epidemiology

WPW syndrome is a disorder characterized by the presence of at least one accessory pathway (AP) that can predispose people to atrial/ventricular tachyarrhythmias and even sudden cardiac death. This diagnosis is reserved only for patients who have pre-excitation on the baseline electrocardiogram (ECG) and symptomatic tachyarrhythmias. In the absence of symptoms, it is preferred the term Wolff–Parkinson–White pattern. The pattern is found in 0.15% to 0.25% of the population, and one-third of these people are thought to develop arrhythmias during a 10-year follow-up [1,2,3,4]. Incidence of life-threatening events (LTE), including sudden cardiac death/sudden cardiac arrest, is not at all trivial, especially in children, reaching 0.8 to 1.9 per 1000 person-years [5].

It is the second most common cause of paroxysmal supraventricular tachycardia in most parts of the world. In large-scale general population studies involving children and adults, the prevalence of WPW syndrome is estimated to be 1–3 in 1000 individuals [6]. A higher prevalence of 0.55% has been reported in first-degree relatives of patients with accessory pathways [7]. Wolff–Parkinson–White syndrome is more commonly diagnosed in men than in women, although this sex difference is not observed in children. Among those with the WPW syndrome, 3.4 percent have first-degree relatives with pre-excitation [8]. In the familial form, a rare early-onset autosomal dominant disease with complete penetrance and variable degrees of expression produced by mutations of the PRKAG2 gene, pre-excitation is associated with ventricular walls thickening due to increased intracellular glycogen deposition in myocytes [7,9,10,11].

Most patients with Wolff–Parkinson–White syndrome have normal anatomy, but some have concomitant congenital heart disease or multisystem diseases (Table 1). Approximately 10 percent of patients with Ebstein’s anomaly have Wolff–Parkinson–White syndrome [11,12,13]. Other congenital heart diseases associated with this syndrome include atrial and ventricular septal defects, coronary sinus diverticula, and corrected transposition of the great vessels [11]. Hypertrophic cardiomyopathy may be associated with WPW, often in the setting of specific gene mutations [14,15,16]. Uncommonly, accessory pathways have also been diagnosed in patients with cardiac rhabdomyoma [12] and X-linked or autosomal recessive HCM phenocopies like Danon, Fabry, and Pompe disease [17].

Multiple APs occur in less than 12% of patients with pre-excitation [13] and in 9% of the pediatric population [18]. In general, they are more common in structural heart disease patients, approximately 50% in patients with Ebstein’s anomaly [19]. The presence of multiple APs can result in a higher risk of supraventricular tachycardia [20], a higher incidence of antidromic re-entry, the potential for more rapid conduction during atrial fibrillation [18,21,22], and ventricular fibrillation. They were found in any combination of pathway locations but with a higher incidence of right free wall and posteroseptal pathways [18,23,24,25]. This may occur due to the presence of discontinuities in the tricuspid annulus fibrosis. 

## 3. Pathophysiology

APs are aberrant muscle bundles that connect the atrium to the ventricle outside the regular atrioventricular conduction system. They are embryologic remnants due to incomplete embryological development of the atrioventricular (AV) annuli without complete separation of the atria and ventricles [25]. 

Up to this point, only a limited number of genes have been identified as potential causes of WPW syndrome; further research in this area is needed. The genetic foundation of the syndrome remains poorly understood, particularly considering its incomplete penetrance and undetermined inheritance patterns in the majority of individuals [26]. Studies [26] have even identified an increased burden of rare deleterious alleles in genes associated with AF in WPW syndrome. While certain genes such as ANK2, NEBL, PITX2, and PRDM16 have been linked to AF/cardiomyopathy in the context of WPW syndrome, PRKAG2 and MYH7 are well-established connections in both sporadic and familial cases. 

ANK2 is responsible for encoding ankyrin-B and plays crucial roles in anchoring and stabilizing multiple ion channels within the cardiomyocyte membrane [27]. On the other hand, PRKAG2 is the most extensively studied gene associated with a familial form of WPW syndrome. Missense variants in this gene are known to activate AMP-activated [27] protein kinase (AMPK), leading to hypertrophic cardiomyopathy (HCM) characterized by glycogen accumulation and ventricular pre-excitation [28].

Various other protein and receptor dysfunctions have been associated with not only the discontinuity of the annulus but also with the gap being bypassed by fast-conducting tissue (Bmp2, Alk3, TBX2, periostin) [29]. This hypothesis is strongly supported by the diagnosis of supraventricular tachycardias (SVT) in utero and by the greater prevalence of WPW syndrome in newborns and infants [30,31]. While there are different types of APs, the most common are short bypass fibers along the mitral or tricuspid annulus.

Gollob et al. [9] suggested that a molecular defect consisting of the substitution of glutamine for arginine (R302Q) may in some way inhibit the normal regression of muscle fibers during atrioventricular septation. Although the propensity for arrhythmias in patients with familial Wolff–Parkinson–White syndrome is well known, the mechanism that triggers these episodes at the molecular level is not understood. The activation of AMP-activated protein kinase in response to beta-adrenergic stimulation could account for the development of tachyarrhythmias during exercise or metabolic stress. Spontaneous automaticity of the APs before or after catheter ablation has also been documented [32,33].

Most of the pathways identified through the use of microscopy have been working myocardium, with only a few reported to contain histologically specialized cells [34]. Whether comprising working myocardium or abnormal myocytes, these pathways have normal gap junctions with a pattern suggestive of working ventricular myocardium [35]. Their electrophysiological features are different from those of the AV nodal conduction system. They typically exhibit fast, nondecremental conduction dependent on a sodium current, very similar to the one found in normal myocardial cells.

Conduction through the bypass tracts can be anterograde, retrograde, or both. Manifest accessory pathways usually conduct in both anterograde and retrograde directions. The degree of pre-excitation is determined by the relative conduction to the ventricle over the AV node (AVN)—His bundle axis versus the accessory pathway (Figure 1) [13]. The majority of anterograde-conducting accessory pathways conduct in both ways. Only a few, less than 10%, have strictly anterogradely conduction, whereas retrograde conduction is met more frequently (~50%). A minority of overt and concealed pathways may present decremental conduction properties [36].

## 4. Diagnosis

The diagnosis of WPW syndrome is reserved strictly for patients who have both pre-excitation and symptoms. Identification of WPW patterns in the general population is extremely difficult, as these patients are, by definition, without palpitations, syncope, or other symptoms secondary to ventricular pre-excitation [37].

The main electrocardiographic features of pre-excitation are short PR interval (<0.12 s), prolonged QRS complex (>0.12 s), and slurred, slow-rising onset of the QRS complex, known as delta wave. Various degrees of pre-excitation are possible, depending on the location of the AP and as well as on AVN conduction properties. Based on the surface ECG it can be also predicted the localization of the manifest accessory pathway. Although the pathway localization and the degree of pre-excitation do not predict the clinical course, they might be important for the risk of abnormal conduction-induced cardiomyopathy [38] and are valuable when considering a catheter ablation procedure. Pre-excitation on the surface ECG can be intermittent, and it is suggested that they may even disappear permanently, especially in neonates (in <35% of cases).

Several algorithms (Chern-En Chiang’s, Fitzpatrick’s, Xie’s, St George’s, and Pamdrun’s algorithms) have been used for predicting the accessory pathway location using different electrocardiographic criteria based on the analysis of the delta wave (Table 2) [39,40,41,42]. The algorithm developed by Arruda et al. [43] utilizing the surface ECG has an overall sensitivity of 90% and specificity of 99% and is still the most used one, even in current practice. The predictive accuracy of the algorithms is significantly reduced in the presence of multiple pathways or coexistent structural heart disease that may additionally alter the QRS morphology. More importantly, in children, the degree of pre-excitation is often minimal due to the fast-conducting AVN-His bundle system, and, therefore, the algorithms are even less efficient. However, more recently developed algorithms seem to have superior accuracy (93%) in comparison with traditional ones (84% with Pambrun and 75% with Arruda), and this was also confirmed in children [44].

The clinical presentation of WPW syndrome is generally unspecific, extremely variable, and, most importantly, it usually accompanies arrhythmic episodes. It is estimated that 90% of children, approximately 65% of adolescents, and 40% of individuals over 30 years with a WPW pattern on the resting ECG are completely asymptomatic [1,11,45]. It should also be noted that the absence of symptoms can be only a temporary condition. On the other hand, there is evidence that in the first year of life, the accessory pathway loses anterograde conduction in as many as 40% of patients [11], and SVT becomes noninducible in a similar percentage, suggesting loss of retrograde conduction [46]. However, this should be interpreted with caution, especially in children and adolescents, since anterograde conduction over APs can only be hidden by augmented AVN conduction (which may leave patients with a fusion narrow QRS complex and a borderline or short PR interval). Children’s electrophysiological results should be interpreted with caution since most studies are performed under sedation/general anesthesia, which may result in decreased AP conduction characteristics [47]. During invasive testing, over 1/3 of children who experience life-threatening arrhythmic events appear to have APs that lack high-risk features because of confounding factors, such as contusion of the AP during catheter insertion and manipulation and variation in autonomic tone [48].

In a normal heart population, the onset of symptoms is age-dependent, and it may vary with accessory pathway location and conduction properties. It may vary from no symptoms at all to slight thoracic pressure or palpitations, or even syncope or sudden cardiac death as the first symptom. Currently, the idea of asymptomatic overt pre-excitation is challenged, as even in children, its presence is associated with reduced exercise capacity (due to APs-induced asynchrony and LV dysfunction) [49]. This might explain the increased risk for heart failure in adults with overt pre-excitation [50]. However, the most worrisome symptoms of WPW are syncope and cardiac arrest. In a long-term prospective study, Munger et al. [1] reported that the onset of symptoms generally happens at a mean of 28 years, but they can appear in any stage of life. Moreover, recent pediatric studies suggest a mean age of 14 years for life-threatening events (LTE) [48]. In infancy, the diagnosis can be extremely challenging because of the nonspecific symptoms such as tachypnea, difficulty in feeding, irritability, abdominal pain, nausea, vomiting, and findings of heart failure if tachycardia continues for a long time [51]. It is important to acknowledge that AVRT can appear during fetal life (most often between 24 and 32 weeks), representing about 70% of arrhythmias during intrauterine life and being one of the most important causes of heart failure with fetal hydrops [29]. Diagnosis is typically made following an episode of atrioventricular reciprocating tachycardia. In a study by Gilljam et al. [52], from 109 patients analyzed with SVT in the neonatal period, 52 of them presented with heart failure, results that emphasize the difficulty in diagnosing an SVT in infants before heart failure develops. Although episodes of SVT often decrease in frequency in the first year of life (~90% of patients) [11], tachycardia recurs in approximately 30% at an average age of 7–8 years [12].
diagnostics-14-00296-t002_Table 2Table 2ECG algorithms for localization of accessory pathways.ECG AlgorithmsStatistiical AnalysisPROCONSAdult AnalysisPediatric AnalysisNo PatientsMean AgeCASpecificitySensitivityPPVNPVAccuracyNo PatientsMean AgeAccuracyBoersma (2002) [53]NA17313 y0.63uses the surface ECGmodest accuracydesigned for childrenreasonable sensitivity and specificity for only five AP-sitesLI (2019) [54]NA10413.6 ± 3.4 y0.92uses the surface ECGonly retrospective analysiseasy to usecould not absolutely differentiate septal wall from free wall APhigh-risk regions can be identified with high accuracy
Min Baek (2020) [55]NA262.0011.7 y0.82superior to other algorithmsless accuracy in younger patientseasy to use—2 stepsfocused on septal pathwaysuses the surface ECGrequires validation in adult patientsMilstein (1987) [56]14134 ± 21 yLL0.940.880.94NA0.90NAuses the surface ECGbased only on four locations of APPS0.950.910.90NAsimple to applyno data about pediatric populationAS0.990.900.97NA
only retrospective analysisRL0.980.750.62NA

Fitzpatrick (1994) [57]14134 ± 21 yL1.001.001.001.000.68NAuses the surface ECGno data about pediatric population
R0.971.100.981.00


St George (1994) [40]36948 ± 10 yallNANANANA0.93NAuses the surface ECGno data about pediatric populationprospective validationlimited data on multiple APseasy to use—requires only 4 stepslower accuracy in predicting right sided APsChiang (1995) [58]36948 ± 10 yallNANANANA0.93NAuses the surface ECGno data about pediatric populationprospective validationlimited data on multiple APseasy to use—requires only 4 stepslower accuracy in predicting right sided APsd’Avila (1995) [59]140NALL0.990.981.00NA0.576415 y0.58uses the surface ECGonly retrospective analysisLP0.981.000.77high accuracy in pediatric populationlimited data on multiple APsLPS0.990.820.90can be used in computerized systems
PS0.970.870.82

RPS0.950.930.70

RL0.981.000.85

AS1.000.921.00

MD1.000.500.10

Iturralde (1996) [60]10232 ± 12 yLPL0.950.910.930.920.88NAuses the surface ECGno data about pediatric populationRI1.000.841.000.95fast to uselimited data on multiple APsLI0.980.840.670.96accurate
RA0.971.000.671.00

RAS0.960.830.550.99

Arruda (1998) [43]25632 yall0.990.900.930.980.80NAuses the surface ECGno data about pediatric populationaccurate in predicting ablation at sites near the AV node and His bundletime consuminguses the initial forces of preexcitation (initial 20 msec)limited data on multiple APsmay aid selection of patients in whom coronary sinus angiography should be performed
Taguchi (2014) [61]144NAall0.990.930.950.98NANAsimple flowchartno data about pediatric populationprospective validationsmall prospective assessmentuses the surface ECG
Pambrun (2018) [42]207NARA0.990.910.880.990.9NAaccurate and reproductibletime consumingRL1.001.000.851.00uses maximal preexcitationrequires EPSRP0.990.960.870.99
no data about pediatric populationPCS0.990.830.970.97

NHS0.980.780.760.99

DCS0.990.670.710.99

LPS0.970.740.770.97

LPL0.980.920.860.99

LL0.991.000.981.00

Easy-WPW (2023) [44]21132 ± 19 yall0.990.920.960.990.935812 ± 4 y0.88reliablelimited data on multiple APsuses the surface ECG
fast and easy to apply—only 2 or 3 steps
analysis on pediatric population



Symptomatic WPW patients may have two main tachyarrhythmias [62]: orthodromic atrioventricular reentrant tachycardia (oAVRT) and atrial fibrillation. Among patients with WPW syndrome, oAVRT is the most common and, most importantly, benign arrhythmia, accounting for 90–95% of reentrant tachycardias that occur in patients with an accessory pathway. Antidromic AVRT (aAVRT) is documented in only 3–8% of WPW patients, but in 30–60% of cases with aAVRT, multiple APs (manifest or concealed) are detected, which could act or not as the retrograde limb during arrhythmia [13]. Other pre-excited tachycardias can also occur in patients with atrial tachycardia, atrial flutter, or atrioventricular nodal reciprocating tachycardia (AVNRT) having the accessory pathway acting as an innocent bystander. These cases are far rarer (one or two cases out of 100) [63]. Although atrial fibrillation can sometimes be of primary onset, it is more often the result of increased atrial pressure due to reduced ventricular filling time triggered by oAVRT [62] and should not be ignored because, in some patients, it can trigger ventricular fibrillation, leading to sudden cardiac death.

## 5. Risk Stratification

Wolff–Parkinson–White is a well-known cause of sudden cardiac death, and in older studies of young survivors of sudden cardiac arrest who did not have structural heart disease, the syndrome was present in up to 33% of patients [64,65]. However, more recent data showed that WPW syndrome is actually a rare cause of SCD, even in children and the young, accounting for as low as 3.6 per 10 million person-years (or at an unlikely 26 per 10 million-person years if all autopsy-negative SCD would be considered produced by pre-excitation) [66]. In asymptomatic patients, the incidence of SCD is very likely even lower [67,68]. However, sudden death may be the first symptom in patients with undiagnosed and/or asymptomatic pre-excitation syndrome [69]. Timmermans et al. [70] found out that in over 50% of cases, cardiac arrest occurs in subjects who are unaware of their condition [64].

### 5.1. Clinical Evaluation

On clinical evaluation, the high-risk features include male sex, familial WPW syndrome (autosomal dominant, chromosome 7, PRKAG2 gene mutation), WPW pattern detected in the first two decades of life, history of atrial fibrillation and arrhythmic symptoms like syncope, and presence of congenital heart disease, especially, Ebstein’s anomaly. Also, high-risk occupations such as those of pilots, bus drivers, and athletes should be given special priority [6].

All patients with WPW syndrome need to undergo risk stratification, given the risk of sudden death (Figure 2). This can be performed using invasive and noninvasive means. The goal is to assess the anterograde refractory period of the accessory pathway, a surrogate marker for the rate of conduction over the pathway during atrial fibrillation [71]. Risk assessment of patients with asymptomatic pre-excitation has limitations, being dependent on the autonomic tone. More accurate models are needed [13].

### 5.2. Noninvasive Assessment

Noninvasive assessment includes in-depth analysis of baseline ECG, 24-h ECG Holter monitoring, and ECG stress test. Previously, the presence of intermittent pre-excitation was considered suggestive of a low risk of SCD (especially sudden disappearance of pre-excitation during exercise) [72]; however, recent data in children and young revealed that at least 13% of these patients still have high-risk APs [13,73]. Catecholamine infusion results in shortening of the anterograde refractory period of the accessory pathway [74] and may unveil high-risk APs in 50% of patients with loss of pre-excitation during exercise [75]. Moreover, there seems to be no clinical, demographical, or electrophysiological difference (including the incidence of high-risk feature APs after isoproterenol testing) between patients with intermittent vs. persistent pre-excitation [76]. Additionally, septal accessory pathways are considered to be at a higher risk of developing ventricular fibrillation [13].

The loss of pre-excitation after administration of the antiarrhythmic drug was used for the identification of a lower-risk subgroup. When, during sinus rhythm, the intravenous injection of ajmaline (1 mg/kg body weight given in 3 min) [77] or procainamide (10 mg/kg body weight over 5 min) [74] results in a complete block of the AP, a long anterograde RP (>270 ms) of the pathway is likely. Noninvasive tests are considered inferior to invasive electrophysiological assessment of the risk of sudden cardiac death, and in patients considered high-risk, invasive assessment should be performed. Currently, this might be useful only in patients who are planning to be temporarily managed with antiarrhythmic drugs (AADs).

### 5.3. Invasive Assessment

Electrophysiological testing (EPT) is recommended for symptomatic patients to elucidate the pathophysiological basis of their arrhythmias and for asymptomatic individuals with high-risk occupations and/or high-risk features, as suggested by noninvasive tests (like multiple APs) [78]. Currently, it also has a class IIa indication, even in asymptomatic patients without a high-risk occupation [13]. This is supported by the fact that asymptomatic and symptomatic patients have similar SCD risks [78,79].

Multiple studies have reported that approximately 20% of asymptomatic patients will demonstrate a rapid ventricular rate during induced AF. During follow-up, however, not all patients will develop symptomatic arrhythmias, and only a minority are exposed to LTE. Although SCD/SCA used to be considered exceptionally rare in asymptomatic subjects [80], more recent multicenter data showed that, albeit small, the risk of SCD/SCA is significantly and unacceptably higher [48], particularly in children, in whom it can be between 0.7% [5] up to 2% for a mean follow-up of 8 years [81]. Although EPT is an imperfect tool, its negative predictive value is considered very good in asymptomatic subjects, especially if performed with catecholamine [71] infusion and without general anesthesia [82].

The most important parameters to be determined during EPT on baseline and (ideally) during catecholamine infusion are those estimating the anterograde conduction over the APs (and therefore the risk for VF), namely the shortest pre-excited RR interval in atrial fibrillation (SPERRI), the accessory pathway effective refractory period (APERP) and the shortest pre-excited paced cycle length during atrial pacing (SPPCL). Inducibility (and ideally the hemodynamic tolerability) for sustained arrhythmia like oAVRT and aVRT, as well as AF, is also important since these are the usual precipitating causes for VF/SCA. The presence of multiple accessory pathways [83] and possible septal ones [12] was also identified as a predictor of future arrhythmic events [12].

Based on combined pediatric and adult studies (age range 5–68 years; mean 28 years), the APERP is less useful in evaluating the risk of sudden cardiac death [70,84]. Assessing the shortest pre-excited RR interval (SPERRI) during AF is the best discriminator for those at risk of VF [12]. Therefore, it is stated that the risk of sudden cardiac death is high when the shortest pre-excited RR interval is <250 ms in the control state in adults, <220 ms in children, or <200 ms during isoproterenol infusion [71,85]. More recently, SPPCL during atrial pacing has shown a good correlation with SPERRI at baseline as well as during isoproterenol infusion, and it can be useful in patients in whom sustained AF cannot be induced [86].

Teo [87] reported that the presence of multiple accessory pathways in conjunction with a SPERRI of less than 250 ms achieved a specificity of 92% and a positive predictive value for future arrhythmic events in 22%. A Bromberg et al. [88] study suggested that a SPERRI of less than 220 ms carried a threefold increase in the risk of SCD compared to the general WPW population.

Regarding the pediatric population, several guidelines defining an approach to asymptomatic WPW have been published. However, more recent data about children with a life-threatening event as the sentinel symptom raise concern, and correct identification and treatment of this population can be lifesaving [20,37,48]. However, SPERRI and SPPCL during EPT have a weak or no correlation with SPERRI measured during a clinical episode of atrial fibrillation with anterograde conduction (Clinical-SPERRI), at least in children (most probably due to the effect of deep-sedation/general anesthesia +/− mechanical inhibition by the diagnostic catheters) [89]. Accurate risk stratification is extremely important, given the number of children with asymptomatic WPW who may be at risk of sudden death and the availability of catheter ablation, which is a highly successful curative procedure [47].

In a Shwayder et al. study [47], none of the EPS-derived risk stratification values obtained with patients under anesthesia had a strong correlation with Clinical-SPERRI. EP-SPERRI was only moderately correlated with Clinical-SPERRI, and there was no significant correlation with the other EPS-derived accessory pathway functional characteristics. Although EP-SPERRI performs the best of the EPS-derived risk stratification measures, the correlation is modest at best, and 24% of the children in this study would have been defined as low-risk based on the EPS-derived risk stratification values but were proven to be high-risk at the time of a clinical event. There are several reasons why an accessory pathway may behave differently, such as position, autonomic tone, supranormal conduction, and, of course, general anesthesia.

## 6. Sudden Cardiac Death

Despite being quite rare and unusual, sudden death may be the first sign of preexcitation syndrome, particularly in the pediatric population [51]. This is extremely important, especially in cases of familial WPW syndrome, as it is associated with a higher risk of sudden cardiac death [69,83]. The incidence of sudden cardiac death in patients with WPW syndrome has been estimated to range from 0.15% to 0.39% over 3- to 10-year follow-up [1,83].

Studies of WPW syndrome patients who have experienced a cardiac arrest have retrospectively identified several markers that identify patients at increased risk, such as SPRRI less than 250 ms during spontaneous or induced AF, history of symptomatic tachycardia, multiple accessory pathways (especially septal AP’s), Ebstein’s anomaly, and familial WPW [71,83].

Cardiac arrest in WPW is generally linked to a cascade of events [62]. The prime mover is AVRT, which, in the presence of elevated atrial vulnerability, triggers atrial fibrillation or atrial flutter. There is no doubt that for cardiac arrest to occur, brief refractoriness of the anomalous pathway is the condition sine qua non. Physical activity (through catecholamine increase) not only facilitates the inducibility of AVRT but also can significantly shorten the anterograde effective refractory period of the accessory pathway, increasing heart rate during atrial fibrillation.

Data pooled from the three largest sets of case records [70,84,90], including 63 adult WPW subjects resuscitated after cardiac arrest, reveal the following main findings: cardiac arrest may occur at any age but generally happens around the age of 20–30 years; in 73% of cases it strikes symptomatic subjects; in the latter case, it especially strikes those who have already suffered episodes of atrial fibrillation (up to 70%); and it is the first clinical manifestation in subjects who are asymptomatic or unaware of their anomaly in a mean 27% of cases (12%, 26%, and 53% in the three sets of case records, respectively).

## 7. Management

### 7.1. Asymptomatic Patients

Management of asymptomatic patients with WPW patterns has always been controversial because asymptomatic does not preclude sudden cardiac death. Up to this point, a clear strategy for these patients has not been established [13]. Most patients will go through life without any clinical events related to their ventricular pre-excitation. Approximately one in five patients will develop an arrhythmia related to their AP during follow-up. The most common arrhythmia in these patients is AVRT (80%), followed by a 20–30% incidence of AF. They generally do not have indications for pharmacological treatment, but they still require risk stratification for sudden cardiac death. Although small, there is present the risk of SCD, which is believed to be higher in younger patients and males [91].

Depending on the noninvasive and/or invasive evaluation, there are several indications in which catheter ablation is indicated, even in asymptomatic patients. The definition of a high-risk pathway is based on demonstrating rapid conduction over the pathway in supraventricular arrhythmia or during atrial pacing maneuvers [84]. The consensus is that if SPERRI during AF is ≤250 ms, SPPCL ≤ 250 ms, and APERP during programmed atrial stimulation is ≤240 ms, then prophylactic catheter ablation of the accessory pathway is recommended [92]. Other recognized risk factors for SCD are younger age, multiple pathways, inducibility of the AVRT during EPS, and septal location of the accessory pathways [13,93]. Also, catheter ablation of the accessory pathway is reasonable in asymptomatic patients if the presence of pre-excitation precludes specific employment (such as with pilots or athletes) [94]. In the most recent guidelines [13,93], it is recommended that all patients with a WPW pattern who are willing to participate in competitive sports undergo invasive EPS for risk stratification and ablation if high-risk properties of the pathway are present. For recreational sports activities, the assessment can be performed initially by noninvasive means.

In children younger than 12 years old, the risk of a fatal event seems to be small, and, therefore, a conservative approach is preferred [93]. It is estimated that approximately 65% of adolescents with a Wolff–Parkinson–White (WPW) pattern on a resting ECG are asymptomatic [37]. That does not mean that young patients with a WPW pattern do not experience life-threatening events in the absence of markers of high risk during the electrophysiological study, as shown by Etheridge et al. [48] in a study that compared 96 case subjects with a history of major life-threatening events and 816 completely asymptomatic subjects. Of case subjects, 60 of 86 (69%) had ≥2 EPS risk stratification components performed; 22 of 60 (37%) did not have EPS-determined high-risk characteristics, and 15 of 60 (25%) had neither concerning pathway characteristics nor inducible atrioventricular reciprocating tachycardia. Moreover, Pappone et al. in 2004 [20] reported that for children with asymptomatic Wolff–Parkinson–White syndrome who are at high risk for arrhythmias, the significant and durable benefits of prophylactic catheter ablation of all accessory pathways outweigh the procedural risks; many (44 percent) of the high-risk patients who did not undergo prophylactic ablation had arrhythmic events, including cardiac arrest or sudden death, during the first two years of follow-up.

### 7.2. Symptomatic Patients

Management of symptomatic patients depends on various factors. Currently, the choice to provide RFA or not has reasonably been based on the presence or absence of symptoms rather than on a specific clinical or electrophysiological algorithm to prevent sudden death [81].

Acute treatment of tachycardia associated with WPW is similar to treating PSVT, with a focus on breaking the cyclical transmission of impulses. This is best accomplished by temporarily prolonging the refractory period of the AV node with drugs such as adenosine [95]. However, adenosine should be used with caution (i.e., with an external defibrillator on stand-by!) in patients with known or suspected anterograde conduction over AP because it reduces atrial ERP, and, combined with bradycardia post-PSVT conversion, can induce AF, which might be transmitted over the AP with a rapid ventricular rate. Ibutilide, procainamide, or flecainide, which are capable of slowing the conduction through the pathway, are preferred.

Treatment of AF associated with WPW is completely different [96]. The basic treatment principle in WPW AF is to prolong the anterograde refractory period of the accessory pathway relative to the AV node to slow the rate of impulse transmission through the accessory pathway and, thus, the ventricular rate. However, in cases with hemodynamic collapse, immediate cardioversion with external electrical shock is mandatory.

RFA has completely revolutionized the approach to the management of WPW syndrome, becoming the method of choice potentially available to all WPW patients. Long-term results from registry studies [81,97,98] demonstrated that there is a striking difference in outcomes between ablated and nonablated patients. Success rates were high, and complications were rare. During follow-up, no ablated symptomatic or asymptomatic patients experienced malignant arrhythmias or VF. Of note, the very high success rates after RFA, as observed by Pappone et al., were associated with very low rates of minor complications (<2%). Even more recently, a large registry from Germany [99] reported mostly complications that have a minor impact on long-term quality of life. A recent meta-analysis showed that an RF catheter ablation of WPW has a >94% success rate, a recurrence rate of 6.2%, with a complication rate of 1% [100]. The widespread utilization of a 3D electro-anatomical system to assist AP RFA, as well as the use of cryo-energy ablation for at-risk septal AP, has led to even higher success combined with reduced recurrences and complications rate, as well as (near) zero fluoroscopic exposure, which is relevant, especially in the pediatric population [101]. In addition to children, good adult candidates for a 3D-guided RFA would be patients at higher complication risk (multiple procedures, septal pathway location, epicardial posteroseptal APs) and those with fragile conduction over APs (spontaneous intermittent pre-excitation, prone to mechanical inhibition by catheters, in case of prior antiarrhythmic medication).

In the long term, successful WPW ablation is associated with reduced mortality (due to SCD prevention, as well as reduced risk of heart failure) [102]. WPW is associated with an increased long-term risk of AF that is not reduced by catheter ablation [102,103], suggesting that genetic factors that determine the development of accessory pathways may be linked to an increased susceptibility of atrial muscle to AF in a subset of patients [26]. However, it seems that the long-term AF risk associated with WPW is significantly reduced if ablation is performed below 50 years of age [102]. Additionally, one group of authors suggested an increased risk of coronary events after WPW RFA, suggesting that RFA in close proximity to coronary arteries might promote accelerated atherosclerosis [100].

## 8. Conclusions

Although it is a pathology rather rare and benign, patients with WPW syndrome should always be assessed for high-risk criteria. Although the lifetime risk of sudden death in WPW syndrome is low, it is “front-loaded” in the young, where sudden cardiac death can be the first symptom. Radiofrequency ablation has drastically changed the management of pre-excitation syndromes. Taking into account the high success rate and low incidence of complications, mostly with low impact on quality of life, this curative procedure should be considered, even in asymptomatic patients.

## Figures and Tables

**Figure 1 diagnostics-14-00296-f001:**
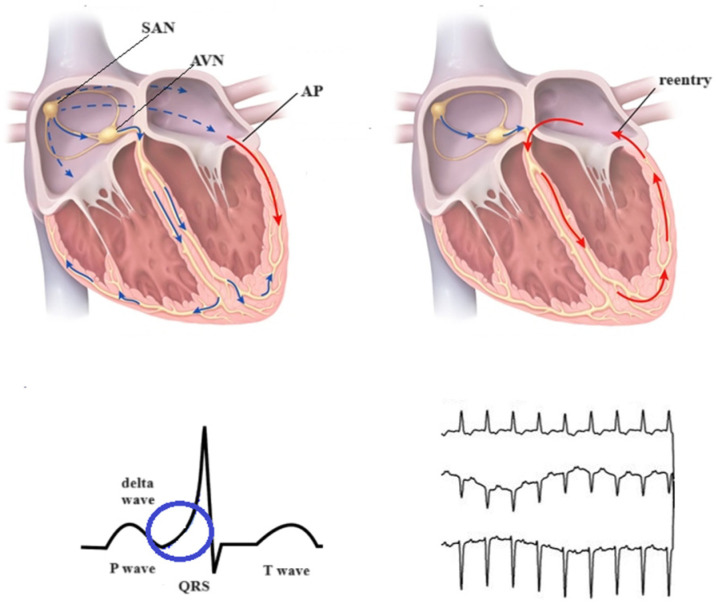
Atrioventricular conduction in ventricular preexcitation. blue arrows = propagation of the electric impulse through the atria and ventricles; red arrow = accessory pathway; blue circle emphasizes the delta wave.

**Figure 2 diagnostics-14-00296-f002:**
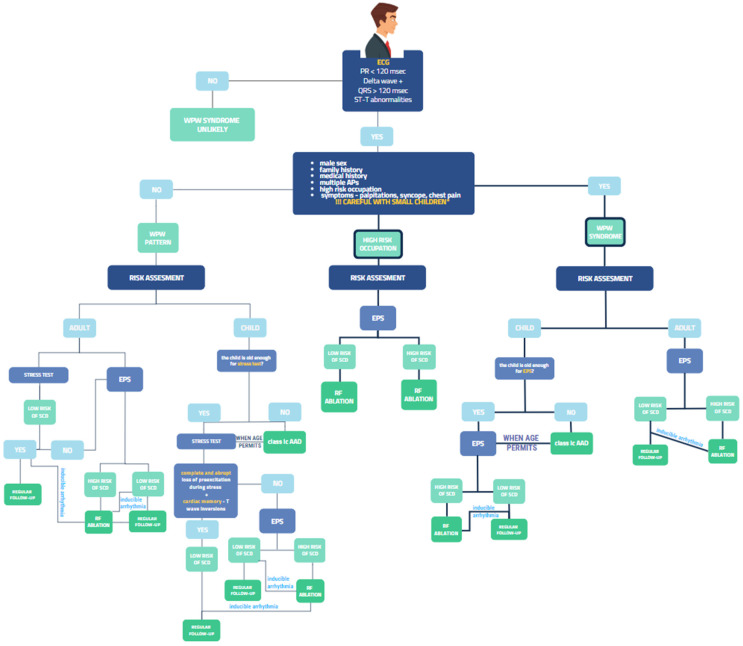
Proposed algorithm for management of patients with ventricular pre-excitation. WPW pattern—only ECG pattern; WPW syndrome—ECG pattern associated with symptoms/tachyarrhythmias; high-risk occupation—e.g., pilots, professional drivers, professional drivers; * children under 25 kg.

**Table 1 diagnostics-14-00296-t001:** The most common conditions associated with WPW syndrome.

Category	Condition	Percent (%)
Congenital heart diseases	tricuspid atresia	0.29–1.3%
	Ebstein anomaly	5–25%
	tetralogy of Fallot	
	pulmonary stenosis	
	coronary abnormalities	
	coronary sinus diverticula	3.6–20%
	corrected transposition of the great vessels	2–5%
	atrial/ventricular septal defects	
	hypertrophic cardiomyopathy	5%
	mitral valve prolapse	
	fibroelastosis	
Multisystem diseases	PRKAG2 syndrome hypokalemic periodic paralysis	
	Pompe disease	
	Danon disease	<1%
	tuberous sclerosis complex; cardiac rhabdomyoma	
Surgically acquired	Corrective surgeries; e.g., Fontan procedure	
	Heart transplantation	
	Valve surgeries	
	Intra-atrial baffles	

## Data Availability

No new data were created or analyzed in this study. Data sharing is not applicable to this article.

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
