# Peer review of "Wolf–Parkinson–White Syndrome: Diagnosis, Risk Assessment, and Therapy—An Update"

_diagnostics, 2024, doi:10.3390/diagnostics14030296_

Round 1

Reviewer 1 Report

Comments and Suggestions for Authors

Dear Authors,

your review “WPW syndrome: diagnosis, risk assessment and therapy – state of the art” is well written and quite informative.

The structure is logical and most of the explanations are easy to understand.

Some additions would really help to better understand the content.

a) could you add a pathophysiological schematic drawing of the heart and where the conduction occurs as figure 1 between pathophysiology and diagnosis?

b) In the pathophysiology section (lines 93ff), more details about the known proteins involved would help the interested reader.

c) in diagnosis line 147: the sentence "...is still the most used..." suggests an older algo, but ref 42 is 2023; please update this and include the year of publication in the text; this would fit very well and increase understanding.

d) Figure 2 is problematic/confusing. The origin of ECG is fine; here you describe wpw pattern. You now make three branches, the left one "wpw pattern" seems to be already included in the first box (ECG). It would be easier to understand if you called the left box "just ECG wpw pattern". The middle and right box, high-risk occupation and wpw syndrome, should be distinguished either in the text or in the caption. in the "high-risk box", the text "!!! careful with small children" also needs further explanation: age period and what to be careful about, this can be added to the caption, e.g. per "*". 

In general, it would be better to make the "YES" branches the most important ones.

The figure is labeled "Risk management for patients with wpw syndrome". However, only the right branch covers patients with wpw syndrome. This is misleading, perhaps "Management of patients with ECG wpw pattern" would be more appropriate.

e) Table 1 is hard to grasp. a lot of work. maybe it would help to put the pediatric on one line with the others and indicate these studies on a separate line.

Comments on the Quality of English Language

English fine so far

Author Response

I trust this message finds you in good health. I am writing to extend my sincere appreciation for your dedicated effort in reviewing my manuscript, titled "WPW syndrome: diagnosis, risk assessment and therapy - state of the art," submitted to Diagnostics. Your insightful feedback has been instrumental in enhancing the manuscript's quality, and I am truly grateful for the thoughtful perspectives you provided.

Having meticulously considered each of your comments and suggestions, I am pleased to share the key revisions made in response to your invaluable feedback:

  1. Pathophysiological Schematic Drawing: Following your guidance, we have incorporated a schematic representation illustrating the heart's anatomy and the conduction system in patients with ventricular preexcitation.
  2. Details about Proteins Involved: To enhance clarity, we have provided additional details regarding the known proteins associated with WPW syndrome.
  3. Correction of Citation: The citation issue regarding "...is still the most used..." has been rectified as per your suggestion.
  4. Figure 2 Modifications: We have updated Figure 2 to include specific details about what is noted with WPW pattern and WPW syndrome. Additionally, the figure's caption now encompasses information about high-risk occupations, and the title has been revised to "Management of patients with ventricular preexcitation" with highlighted important branches.
  5. Table 1 Modifications: Changes have been implemented in Table 1 to enhance its readability and overall user-friendliness.

I firmly believe that these revisions have significantly fortified the manuscript, addressing the concerns raised during the review process and improving clarity, accuracy, and overall content quality.

Your thorough review reflects your commitment to upholding the high standards of Diagnostics, and for that, I am truly grateful. Should there be any lingering concerns or areas requiring further attention, please feel free to bring them to my attention.

Thank you once again for your time, expertise, and invaluable feedback. I am hopeful that my revised manuscript will be favorably considered for publication in Diagnostics.

Reviewer 2 Report

Comments and Suggestions for Authors

General comments:

The manuscript presented by the authors present a systemic review for WPW. The novelty of the study is average. There are minor concerns which need to be addressed before publication. Specific comments follow:

1.     The title, objectives and the contents of this review did not fully match with each other. While the authors said it is “state of the art” review, there were great number of contents is a history recap and relatively old references of the WPW syndrome. A clear and corresponding description should be specified, and the references of the novel contents should be up to date.

2.     Sub-heading may help with a better presentation while some of the paragraphs were too long and with complicated syntax.

3.     Review could include authors interpretation of the management, and integrate the current knowledge gap in WPW to improve the scientific soundness.

Author Response

I trust this correspondence finds you in good health. I am reaching out to express my sincere gratitude for your dedicated efforts in reviewing my manuscript, titled "WPW syndrome: diagnosis, risk assessment, and therapy - state of the art," submitted to Diagnostics. Your insightful feedback has played a pivotal role in elevating the manuscript's quality, and I am genuinely appreciative of the thoughtful perspectives you offered.

After thoroughly considering each of your comments and suggestions, I am pleased to outline the key revisions made in response to your invaluable feedback:

  1. The title has been modified accordingly. It is true that we included some old references, however those are important steppingstone studies that are cited even in the most recent guidelines. Moreover, we added a lot of quite new and important studies (some of them 2023!) that are clarifying some controversial issues (like spontaneous loss of conduction properties of accessory pathways or limits of invasive EP testing) and/or add new perspectives on effects of ventricular pre-excitation (like LV dysfunction/heart failure/AF risk). We respectfully ask the reviewer to verify this in the references list.
  2. We introduced subheadings, thank you for your suggestion.
  3. We respectfully remind the reviewer that this is our work and therefore represents our interpretation and experience (for instance see fig 2). We avoided to cite directly our (unpublished yet) experience and we cited scientifically sound published data (Circulation, Europace).

Your meticulous review underscores your commitment to maintaining the high standards of Diagnostics, for which I am truly grateful. Should there be any lingering concerns or areas necessitating further attention, please do not hesitate to bring them to my attention.

Thank you once again for investing your time, expertise, and providing invaluable feedback. I am optimistic that my revised manuscript will be favorably considered for publication in Diagnostics.